# Prevalence of ESBL, AmpC and Carbapenemase-Producing Enterobacterales Isolated from Raw Vegetables Retailed in Romania

**DOI:** 10.3390/foods9121726

**Published:** 2020-11-24

**Authors:** Ioana Alina Colosi, Alina Mihaela Baciu, Răzvan Vlad Opriș, Loredana Peca, Tristan Gudat, Laura Mihaela Simon, Horațiu Alexandru Colosi, Carmen Costache

**Affiliations:** 1Department of Molecular Sciences, Division of Microbiology, Iuliu Hațieganu University of Medicine and Pharmacy, 6 Louis Pasteur Street, 400349 Cluj-Napoca, Romania; icolosi@umfcluj.ro (I.A.C.); alinabacium@gmail.com (A.M.B.); tristan.gudat@web.de (T.G.); lauramihaelasimon@yahoo.com (L.M.S.); carmen_costache@yahoo.com (C.C.); 2Department of Molecular Sciences, Division of Medical Genetics, Iuliu Hațieganu University of Medicine and Pharmacy, 6 Louis Pasteur Street, 400349 Cluj-Napoca, Romania; danapeca@gmail.com; 3Department of Medical Education, Division of Medical Informatics and Biostatistics, Iuliu Hațieganu University of Medicine and Pharmacy, 6 Louis Pasteur Street, 400349 Cluj-Napoca, Romania; hcolosi@umfcluj.ro

**Keywords:** ESBL production, AmpC β-lactamase, carbapenemase, Enterobacterales, fresh vegetables, household washing, food safety

## Abstract

(1) Background: As β-lactamase-producing Enterobacterales are no longer exclusively associated with the health care system, investigating the potential risk they pose to the integrity of the environment and food safety has become of utmost importance. This study aimed to determine the prevalence of extended-spectrum β-lactamase (ESBL), AmpC, and carbapenemase-producing Enterobacterales isolates from retailed raw vegetables and to determine if household washing is an effective method of lowering bacterial load; (2) Methods: Seasonal vegetables (*n* = 165) were acquired from supermarkets (*n* = 2) and farmer markets (*n* = 2) in Romania. Following sample processing and isolation, identification of Enterobacterales was performed by matrix-assisted laser desorption ionization time-of-flight mass spectrometry (MALDI-TOF). Polymerase chain reaction (PCR) multiplex was used to ascertain the presence of the main ESBL, AmpC, and Carbapenemase genes. Phenotypic antibiotic resistance profiles of isolates were determined by extended antibiograms. *Enterobacteriaceae* colony-forming units (CFU) counts were compared between vegetable types; (3) Results: Beta-lactamase producing bacteria were observed on 7.9% of vegetables, with 5.5% displaying ESBL/AmpC phenotype and 2.4% identified as Carbapenemase producers. The most frequently detected β-lactamase genes were *bla*_SHV_ (*n* = 4), followed by *bla*_CTX-M_ and *bla*_TEM_ (each with *n* = 3). Phenotypic antibiotic resistance analysis showed that 46% of isolates were multiple drug resistant, with aminoglycosides (38.5%) the most prevalent non-β-lactam resistance, followed by first-generation quinolones (38.5%). (4) Conclusions: The present study has described for the first time the presence of β-lactamase producing Enterobacterales in fresh produce retailed in Romania.

## 1. Introduction

The “antibiotic era”, ushered in by Paul Ehrlich and Alexander Fleming, marked the beginning of a golden age for medicine. Since then, antibiotics, their development, and use, have become the cornerstone of modern medical practices around the globe [1]. However, due to the extensive use of antibiotics in human and veterinary medicine, aquaculture, and agriculture, antibiotic-resistant bacteria (ARBs) have emerged at an alarming rate [2]. While in past times, antimicrobial resistance was associated with hospital settings and medical care, nowadays, the dissemination of ARBs in non-medical environments is recognized and identified as an evolving problem. Information regarding the prevalence of ARBs in primary food production is an essential element required to adequately determine human exposure [3].

Due to the intense use of antibiotics in agriculture, fresh produce that are often consumed raw, such as fruits and vegetables, have been recognized as vectors for the transmission of pathogenic bacteria and ARBs [4]. It is important to note that edible plants and fruits can become contaminated with both antibiotic-resistant and pathogenic bacteria during growth, through the use of animal biofertilizers, such as wastewater and manure [5]. In order to limit excessive antibiotic use in the livestock industries, in 2003, the European Parliament instituted Regulation (EC) No. 1831/2003 on animal nutrition additives which forbids the use of antibiotics as growth boosters in European countries [6]. Thus, veterinary use of antibiotics is limited to therapeutic and prophylactic practices. Nevertheless, in 2016, a total of 7787.1 tonnes of antibiotic active ingredients were purchased for use on food-producing animals in 30 European countries. The largest amounts of antibiotic were tetracyclines (32%), penicillins (26%), and sulfonamides (12%). Although this amount is 15% lower than that reported in 2010, it remains alarmingly high [7]. In addition, contamination of fresh produce can occur during harvesting, distribution, and in stores, as a result of non-hygienic human practices and incorrect handling [8].

Human infection with pathogenic bacteria is generally well understood and surveyed during outbreaks by linking the pathogen to the source. As such, numerous outbreaks with pathogenic bacteria have been associated with the consumption of vegetables. Between 2015 and 2018 an outbreak of *Listeria monocytogenes* that affected Austria, Denmark, Finland, Sweden, and the United Kingdom was attributed to contaminated frozen vegetables [9]. In the USA, a recent (June 2020) outbreak of *Salmonella* Newport that affected 48 states (1127 cases, 167 hospitalizations) was linked to the consumption of red onions [10].

However, the primary source and transmission pathways of antibiotic resistance genes remain a point of debate within the medical community. Extended-spectrum β-lactamases (ESBLs), AmpC β-lactamases (AmpC), and carbapenemases have been increasingly reported in gram-negative bacteria worldwide [11]. Cases of human carriage and infection with a wide range of *Enterobacteriaceae* carrying β-lactamase resistant genes have been reported in patients that had not received any prior medical care; a phenomenon observed even in countries with low antibiotic consumption [12]. Due to the constant increase in human infections, the Center for Disease Control and Prevention has catalogued ESBL and carbapenemase producing *Enterobacteriaceae* as an urgent threat to public health as resistance to β-lactams, which are often first-line treatment options, significantly limit the number of available effective therapies [11]. Furthermore, infections with β-lactamase-producing bacteria can lead to delays in the commencement of appropriate antimicrobial treatment, prolonged hospital stay, increased mortality, and morbidity [13]. Enterobacterales harboring antibiotic resistance genes have been isolated from livestock and animal retail foods in European countries [14].

Although a number of studies have highlighted the presence of ESBL/AmpC-producing bacteria on vegetables [15,16], the prevalence of β-lactamase-producing Enterobacterales on fresh vegetables has not been documented in Romania. This is of significance as Romania has one of the highest levels of antimicrobial resistance among European countries. According to surveillance reports published by the European Antimicrobial Resistance Surveillance Network (EARS-net), in 2015, 26.8% of *E. coli* and 70.7% of *Klebsiella pneumoniae* strains isolated from patients presented resistance to 3rd-generation cephalosporins [17]. More alarmingly, the percentage of carbapenem-resistant Gram-negative bacteria were especially high, with *K. pneumoniae* and *Acinetobacter baumanii* at 24.7% and 81.5%, respectively [18]. This, coupled with the fact that a Romanian adult consumes on a daily basis an average of 325 g of vegetables and vegetable products [19], raises the need to investigate the potential of fresh produce to harbor and disseminate antibiotic resistance genes.

The purpose of the present study was to determine the prevalence of ESBL, AmpC, and carbapenemase-producing Enterobacterales from fresh vegetables in Romania.

## 2. Materials and Methods 

### 2.1. Sample Collection

In order to determine the prevalence of ESBL, AmpC, and carbapenemase-producing Enterobacterales on fresh produce, during April–May of 2019, a total of 165 vegetables were acquired from supermarkets (*n* = 2) and farmer markets (*n* = 2) in the city of Cluj-Napoca, Romania. The 11 types of vegetables that were sampled from each market included: radish (*n* = 15), spring onions (*n* = 15), cabbage (*n* = 15), cucumbers (*n* = 15), cornichons (*n* = 15), lettuce (*n* = 15), spinach (*n* = 15), carrots (*n* = 15), parsley (*n* = 15), peppers (*n* = 15) and, tomatoes (*n* = 15). Only vegetables that were labeled as “grown in Romania” were chosen. Selection of produce was based on the dietary preferences of the Romanian population. Samples that had visible damage were excluded from the study. Vegetable samples were processed within 3 h of their acquisition.

### 2.2. Enterobacteriaceae Colony-Forming Unit Count

Two grams of each vegetable sample (pulp and peel) was mashed with a sterile mortar and pestle. Each mashed sample was added to 18 mL of peptone water. Following this, one µL of the vegetable mixture was diluted in 1 mL sterile saline. One µL of this dilution was then inoculated onto an EBM agar plate and incubated for 24 h at 37 °C. A colony-forming unit (CFU) count was performed by two individual researchers, using the ImageJ software version 1.52 r (Laboratory for Optical and Computational Instrumentation, University of Wisconsin, Madison, USA). The dilution was performed to allow accurate CFU counts on the agar plate.

### 2.3. Isolation and Identification of Enterobacterales

After the sample for the CFU count was taken, peptone-enriched samples were incubated for 24 h at 37 °C. Five µL of the enriched samples were streaked onto Brilliance UTI Agar plates (Oxoid, Johannesburg, SA) in order to allow for morphological and color-coded distinction of individual bacteria according to the specifications of the producer. A maximum of 6 morphological distinct colonies were recovered from each plate, isolated and then purified. Identification of isolates was performed by matrix-assisted laser desorption ionization time-of-flight mass spectrometry (MALDI-TOF) (Bruker, Bremen, Germany). Only identifications that displayed a single result with a confidence score ≥2 were considered acceptable. Non-Enterobacterales isolates were not included in further analysis.

### 2.4. Antimicrobial Susceptibility Testing

In order to determine the resistance patterns of isolates, the Kirby-Bauer disc diffusion technique was used [20]. All isolates were screened for ESBL and AmpC production using the disc diffusion test with cefpodoxime 10 μg, cefotaxime 30 μg, and ceftazidime 30 μg (Oxoid, Johannesburg, SA). Inhibition zone diameters were compared with the EUCAST criteria to determine if isolates were susceptible, intermediate, or resistant. Resistance to cefpodoxime was regarded as a phenotypic indicator of ESBL production, while resistance to ceftazidime or cefotaxime and cefoxitin 10 μg was considered an indicator of AmpC production [20]. ESBL production was confirmed using the combination disc test with cefotaxime, ceftazidime, cefepime 30 μg, and amoxicillin-clavulanic acid (20 μg/10 μg). ESBL confirmation was considered positive if expansion of any of the cephalosporin inhibition zone towards amoxicillin-clavulanic acid disc occurred. Screening for Carbapenemase production in isolates was performed using the disc diffusion method with meropenem 10 μg and imipenem 10 μg, according to the EUCAST methodology [20]. Furthermore, all isolates were tested for susceptibility or resistance to ampicillin (10 μg), amoxicillin-clavulanic acid (20 μg/10 μg), piperacillin-tazobactam (100 μg/10 μg), imipenem (10 μg), meropenem (10 μg), neomycin (10 μg), cefuroxime (30 μg), ceftazidime (30 μg), cefepime (30 μg), cefotaxime (30 μg), cefoxitin (10 μg), ceftriaxone (30 μg), trimethoprim-sulfamethoxazole (1.25 μg/23.75 μg), tetracycline (30 μg), doxycycline (5 μg), gentamycin (10 μg), amikacin (30 μg), chloramphenicol (10 μg), norfloxacin (5 μg), ciprofloxacin (5 μg), levofloxacin (5 μg), nalidixic acid (30 μg) (Oxoid, Johannesburg, SA) in accordance with Clinical and Laboratory Standards Institute (CLSI) guidelines [21]. Isolates resistant to a minimum of three antimicrobial classes were considered multi-drug resistant (MDR). The following bacterial strains were used as positive and negative controls: *E. coli* ATCC 25922, *K. pneumoniae* ATCC 700603, *K. pneumoniae* NCTC 13438 and *K. pneumoniae* CCUG 58545.

Each of the isolates phenotypically identified as ESBL, AmpC, and carbapanemase producers were placed in a BHI (brain heart infusion) broth and allowed to incubate overnight at 37 °C. A 7% solution of dimethyl sulfoxide (DMSO) was added to each liquid culture (1:3 ratio). The isolates were stored at −80 °C until molecular processing (maximum six months) [22].

### 2.5. Genomic DNA Extraction and Polymerase Chain Reaction (PCR) Amplification

PCR multiplex was used to ascertain the presence of the main ESBL, AmpC, and carbapenemase genes. Single colonies of each isolate were cultured in BHI broth for 18 h at 37 °C. The bacterial cells were pelleted by centrifugation (14,500 g for 5 min). DNA extraction was performed using the Wizard^®^ Genomic DNA Purification kit (Promega, Madison, USA), and DNA concentration was determined using the Eppendorf BioPhotometer plus (Eppendorf AG, Hamburg, Germany). Total DNA (1 µL) was subjected to each multiplex PCR in a 25 µL reaction mixture containing 12.5 µL PCR Master Mix (ThermoFisher Scientific, Johannesburg, SA), 1 µL bovine serum albumin (ThermoFisher Scientific, Johannesburg, SA), 8.3 µL specific primer and 2.2 µL nuclease-free water. Amplification was carried out as follows: initial denaturation at 94 °C for 10 min; 30 cycles of 94 °C for 40 s, 60 °C for 40 s and 72 °C for 1 min; and a final elongation step at 72 °C for 7 min [23]. Amplicons were visualized after running at 100 V for 1.5 h on a 2% agarose gel (MetaPhorTM Agarose, Lonza Bioscience, Basel, Switzerland). A 100 bp DNA ladder (ThermoFisher Scientific, Johannesburg, SA) was used as a size marker. Specific primers for each of the genes were ordered from ThermoFisher Scientific and are described in Table 1 [24]. Positive and negative controls were used to test the multiplex PCR protocol Appendix A.

### 2.6. Statistical Analysis

Statistical analysis was performed using GraphPad Prism software version 8.4.3 (GraphPad, CA, USA). Descriptive results were presented as mean and standard deviation (SD). In order to assess the hypothesis that some vegetable types could routinely harbor a more diverse selection of bacteria than others, the average number of distinct bacterial colonies isolated from each vegetable type was determined and compared. To achieve this, one-way ANOVA was used, followed by Tukey’s post-hoc tests for multiple comparisons. Finally, an evaluation of the total *Enterobacteriaceae* CFU count of vegetables in relation to the origin of the sample (farmer’s market vs supermarket) was conducted. This was achieved by applying either a parametric test (unpaired *t*-test) or a non-parametric test (Kolmogorov–Smirnov test), depending on the normal or skewed distribution of the compared series. The threshold for statistical significance was set at α = 0.05.

## 3. Results

### 3.1. Bacterial Diversity

A total number of 6 bacterial colony types were identified on Brilliance UTI agar, based on the instructions provided by the manufacturer. For each vegetable sample, we recorded the number of distinct colony types observed and isolated. Tomato samples showed the lowest amount of contamination, yielding an average of 2.2 distinct colony types, followed closely by bell pepper and spring onion samples with 2.4 as can be noted in Figure 1. On the other hand, gherkin, spinach, and parsley exhibited a more diverse flora, generating an average of 3.9, 4.1, and 4.9 distinct colonies per sample, respectively. High statistical significance was observed when comparing tomatoes to the above mentioned (*p* < 0.0005, *p* < 0.0001, *p* < 0.0001 respectively). Bell peppers and spring onions showed highly similar results (*p* < 0.005, *p* = 0.0001, and *p* < 0.0001 respectively). Overall, leafy vegetables harbored more morphologically distinct bacteria than the other types of fresh produce included in the study.

### 3.2. Enterobacteriaceae Colony-Forming Unit (CFU) Count

All vegetable samples exhibited similar Enterobacteriaceae CFU counts Table 2.

All in all, vegetable samples presented similar Enterobacteriaceae CFU counts whether they were acquired from supermarkets or farmer’s markets Figure 2. No relevant statistical significance was observed, with the exception of tomato samples, which showed higher CFU counts in farmer markets (4.8 log) compared to supermarkets (2.6 log) (*p* < 0.005).

### 3.3. Identification and Prevalence of β-Lactamase-Producing Enterobacterales

Of the 856 bacterial isolates recovered from the vegetable samples, 5 (0.58%) exhibited ESBL production, 4 (0.47%) AmpC and 4 (0.47%) carbapenemase. The following Enterobacterales with resistant phenotype were identified: *Enterobacter cloacae* (*n* = 5), *Enterobacter ludwigii* (*n* = 1), *Citrobacter brakii* (*n* = 1), *Citrobacter freundii* (*n* = 1), *Proteus vulgaris* (*n* = 1), *Seratia marcescens* (*n* = 1), *Escherichia coli* (*n* = 1), *Morganella morganii* (*n* = 1), *Klebsiella oxytoca* (*n* = 1). Moreover, 7.9% (13/165) of fresh produce contained at least one bacterium with an antimicrobial-resistant phenotype, with the following distribution pattern: 20% (3/15) of carrots, 20% (3/15) of spinach, 13.3% (2/15) of cucumber, lettuce and parsley, and finally 6.6% (1/15) of cabbage. Of the 13 vegetables contaminated with antibiotic-resistant-Enterobacterales (AR-E), 61.5% (8/13) were acquired from farmer markets, while 38.5% (5/13) from supermarkets. The bacteria and vegetable from which they were isolated, together with the market from which the vegetable was purchased are presented in Table 3.

### 3.4. Antibiotic Resistance Genotype Profile

Genes encoding β-lactamases were detected in 69.2% (9/13) of isolates obtained from fresh produce. The most frequently detected b-lactamase genes were *bla_SHV_* (*n* = 4), followed by *bla_CTX-M_* (*n* = 3), *bla_TEM_* (*n* = 3), *bla_KPC_* (*n* = 2), *bla _DHA_* (*n* = 1) and *OXA-48* (*n* = 1).Five isolates harbored more than one gene; *E. coli* displayed the *bla*_CTX-M_ gene in association with *bla*_TEM_, two isolates (*S. marcescens* and *P. vulgaris*) carried the *bla_TEM_* and *bla_SHV_* genes, one *E. cloacae* presented *bla_CTX-M_* with *bla_SHV_*, and lastly, *K. oxytoca* with *bla_KPC_* and *bla_SHV_*
Table 3. Despite repetitive PCR assays, in 4 of the samples phenotypically identified as β-lactamase producers, the resistance genes could not be determined.

### 3.5. Antibiotic Susceptibility Patterns

As shown in Table 3, 46% (6/13) of isolates presented MDR profiles. The most prevalent non-β-lactam resistance was against aminoglycosides (9/13 isolates), followed by first-generation quinolones (5/13). The *E. coli* isolate was the only one to present resistance to tetracyclines. Similarly, resistance to sulfonamides was observed in one *E. cloacae* isolate.

## 4. Discussion

The presence of bacteria with increased antimicrobial resistance on fresh vegetables has been receiving increased attention from the medical community, the food industry, and the general public. To our knowledge, there is little to no information regarding the presence of AR-E from raw vegetables retailed in Romania. In order to fill this gap in knowledge, the principal aim of the present study was to detect and record the prevalence of ESBL, AmpC, and carbapenem-producing Enterobacterales in raw vegetables grown and distributed in Romania. To achieve this, 165 vegetable samples (11 different types) purchased from farmer markets and supermarkets in Cluj-Napoca, Romania were analyzed. Beta-lactamase producing bacteria were observed on 7.9% of vegetable samples, with 5.5% displaying ESBL/AmpC phenotype and 2.4% identified as carbapenemase producers. The vegetables that displayed the highest contamination rates with β-lactamase producing bacteria were carrots and spinach (20% of samples contaminated), followed by cucumber, lettuce and parsley (13.3%) and lastly cabbage (6.6%). The percentage of ESBL/AmpC is similar to that observed by Reuland et al. (2014) [23] on retail vegetables from the Netherlands (6%) but considerably lower than the 25.4% reported by Zurfluh et al. (2015) [25] for vegetables imported by Switzerland. Richter et al. (2019) also reported a high prevalence (17.4%) of ESBL/AmpC bacteria isolated from raw vegetables in Gauteng Province, South Africa [26]. Regarding carbapenem-resistant Enterobacterales, Liu et al. (2018) reported a similar prevalence in ready-to-eat vegetables from China, as 10 out of 411 samples analyzed were contaminated [27].

While in the past, studies on fresh produce have identified mainly environmental bacteria carrying chromosomal β-lactamase resistance genes [5,28], recent studies paint a very different picture, with a high prevalence of pathogenic bacteria (*Salmonella* spp.) and opportunistic pathogens such as *E. coli, Klebsiella pneumoniae, Citrobacter* spp. and *Enterobacter* spp. [8,29,30]; bacteria that are not only able to cause community-acquired human infections, but often carry resistance genes located on transmissible plasmids. Similar to recent publications, in the present study the overwhelming majority of β-lactamase producing bacteria were *Enterobacter* spp. (*n* = 6), followed by *Citrobacter* spp. (*n* = 2) and *E. coli, S. marcescens, M. morganii, K. oxytoca, P. vulgaris* (each with *n* = 1). It is important to note that a recent study by Farkas et al. (2019) concerning the antibiotic resistance profile of *Enterobacteriaceae* showed that the most prevalent pathogenic bacteria isolated from humans in the city of Cluj-Napoca, Romania, were *E. coli* and *Enterobacter* spp. [31].

MDR (resistance to ≥3 antimicrobial classes) was identified in 46.1% of isolates. The most prevalent non-β-lactam resistance was against aminoglycosides (69.2%), followed by first-generation quinolones (38.5%). Only one *E. coli* isolate presented resistance to tetracyclines. Similarly, resistance to sulfonamides was observed in one *Enterobacter cloacae* isolate. Ben Said et al. (2016) reported a similar percentage of aminoglycoside resistance (66.7%) [32], while Richter et al. (2019) registered much higher rates, at 96.1% [26].

ESBLs include CTX-M-, TEM-, SHV-, and OXA-type enzymes [33]. Their prevalence and distribution have shifted since the 1980s when 3rd-generation cephalosporin resistance among Enterobacterales was mainly due to TEM- and SHV-type ESBLs [34]. For the last 20 years, CTX-M has dominated SHV and TEM, and is currently the most frequent ESBL worldwide [35]. However, due to the variance in antibiotic consumption, therapies, and strategies employed, the prevalence of ESBLs tends to differ from country to country. In Romania, CTX-M is the most common type of ESBL isolated from humans in clinical settings. Maciuca et al. (2015) also highlighted that *E. coli* isolated from poultry in Romania had the highest international prevalence of CTX-M-15, with 53% of isolates carrying this gene [36]. In the present study, 5 isolates were identified as ESBL producers. Genotypic characterization by PCR assays revealed *bla_SHV_* to be the most frequently isolated ESBL, followed by *bla_CTX-M_* and *bla_TEM_*. This result is quite different from previous studies from other countries that detected either *bla_CTX-M_* [37,38,39,40] or *bla_TEM_* [41,42] as the predominant ESBL gene in their isolates. Although according to phenotype, 4 isolates were identified as AmpC producers, genotyping was successful for only one *E. cloacae* isolate which presented the *bla_DHA_* gene. *Enterobacter* spp. carrying *bla_DHA_* AmpC β-lactamases have been previously described in isolates originating from fresh produce [37] and humans [43]. As previously mentioned, Romania is currently facing a rapid emergence of carbapenem-resistant bacteria, all of the major carbapenemases being previously detected in *K. pneumoniae* isolates collected from Romanian patients [44]. Thus, the present study also analyzed the prevalence of carbapenemase-producing Enterobacterales from fresh produce. A total of 4 isolates were identified as carbapenemase producers; PCR assay genotyping revealed a *K. oxytoca* isolate to harbor *bla*_KPC_, *E. cloacae* with *OXA-48* and *M. morganii* displaying *bla*_KPC_. Previous studies have also isolated carbapenemase-producing *Enterobacteriaceae* from seafood [45], animals [27], retail meat [46] and fresh vegetables [27,47,48].

The presence of bacteria pertaining to the *Enterobacteriaceae* family is used to assess the general hygiene level of various foods. The Health Protection Agency (United Kingdom of Great Britain) has set a guideline for assessing the microbiological safety of ready to eat foods, classifying *Enterobacteriaceae* count in 3 levels: unsatisfactory (>4 log CFU/g), borderline (2–4 CFU/g) and satisfactory (<2 CFU/g) [49]. Although their presence on fresh produce is expected and generally not considered hazardous, high numbers can be an indication of poor practices [49]. The Enterobacteriaceae population levels in the present work ranged from 4.51 log (tomatoes) to 5.51 log (carrot), results which are in accordance with previous publications that have analyzed the microbial load of fresh vegetables [50,51]. While this does not necessarily reflect poorly on the microbiological safety of the vegetables analyzed, it has been previously observed that a high Enterobacteriaceae count is associated with the application of manure and/or wastewater prior to harvesting, and excessive handling during harvest, shipment and distribution [52].

Another aim of this study was to determine if high microbial load in vegetables is correlated with a broader microbial diversity. For this, as previously described, a maximum of 6 morphologically distinct colonies were isolated and purified from each sample. It is important to note that a single parsley sample yielded 6 types of colony. The vegetables that displayed the lowest number of distinct bacteria were tomato (an average of 2.2 distinct colony types), bell pepper (2.4), and spring onion (2.4). Incidentally, these also registered the lowest CFU counts of all vegetable types. This can be attributed to the fact that tomatoes and bell peppers grow above ground and come in contact less often with manure and wastewater that is used as fertilizer, thus limiting their contamination with fecal *Enterobacteriaceae*. In the case of spring onions, it has been previously demonstrated that their juices and vapors are able to inhibit the growth of numerous bacteria including *E. coli*, *Bacillus cereus*, *Staphylococcus aureus*, and *Salmonella* spp. [53]. Leafy and bulb vegetables displayed the highest microbial diversity which was correlated with high CFU counts. In addition to possible contamination with various bacteria found on the soil, due to their structure, leafy vegetables are more vulnerable to microdamage caused by environmental factors (e.g., wind, heavy rainfall). These lesions in turn can act as an entry portal for bacteria that attach preferentially to cut edges [54].

Lastly, we compared vegetables acquired from supermarkets and those from farmers’ markets in order to discern if there are any meaningful differences from a microbiological point of view. Vegetable samples displayed similar microbial loads whether they were acquired from supermarkets or farmers’ markets. The only exception were tomatoes, which displayed an average of 2.2 log decrease in CFU count in samples from supermarkets. This could be attributed to practices that are often employed by supermarkets in order to prolong shelf-life and increase the visual appeal of produce, such as thoroughly washing and applying edible coatings on produce. Tomatoes in retail markets are often waxed in order to prevent external microbial contamination, moisture desorption/absorption, all the while maintaining their organoleptic properties for extended periods of time [55]. Furthermore, various substances that act as natural antimicrobials (such as essential oils) are being incorporated increasingly in edible coatings [56]. Regarding AR-E, 61.5% of isolates originated from farmers’ markets and 38.5% from supermarkets. Given the present results, we could not find any statistically significant difference in the microbiological quality of vegetables available in farmers’ markets and supermarkets.

## 5. Conclusions

The present study has described for the first time the presence of β-lactamase producing Enterobacterales in fresh produce retailed in Romania. The results obtained indicate that further investigation of different vegetable types from other regions of Romania is necessary. These findings suggest that under certain circumstances, produce that is often consumed raw and considered beneficial for the human body, can pose serious health risks. In order to properly ascertain the true impact of the increasing prevalence of antibiotic resistance genes along the food chain, future studies should focus on identifying the effects and occurrence rate of human colonization with ARBs originating from fresh produce.

## Figures and Tables

**Figure 1 foods-09-01726-f001:**
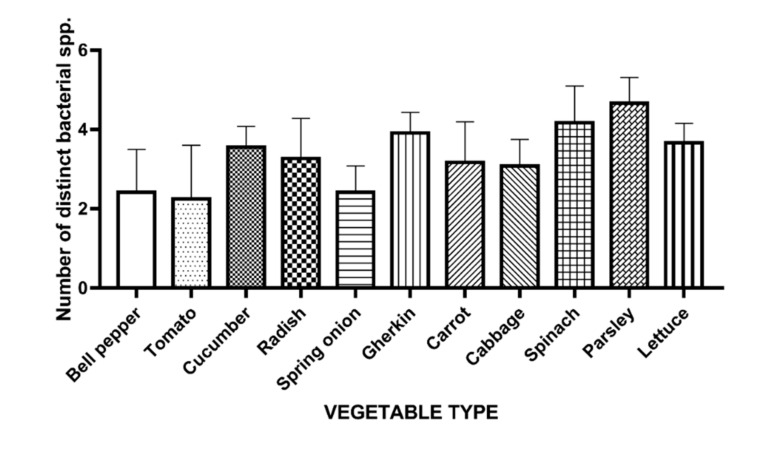
Average number of morphologically distinct Gram-negative bacterial species isolated from each vegetable type.

**Figure 2 foods-09-01726-f002:**
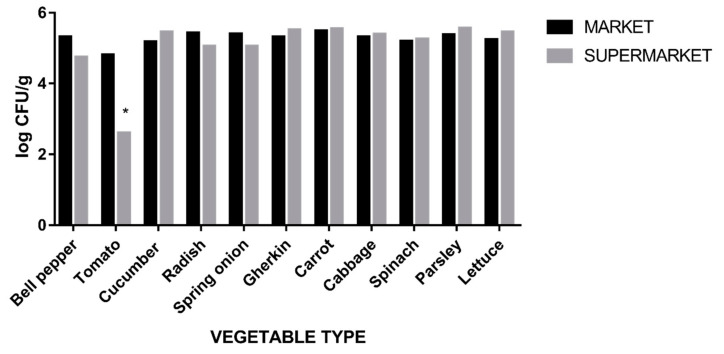
Average *Enterobacteriaceae* CFU counts, comparison between vegetables obtained from farmer markets and supermarkets, * = *p* < 0.05.

**Table 1 foods-09-01726-t001:** Group-specific primers used for polymerase chain reaction (PCR) multiplex assay [24].

Beta-lactamase (s) Targeted	Primer Sequence For/Rev	Length (Bases)	Primer Concentration (pmol/µL)	Amplicon Size (bp)	Annealing Position ^*^
TEM variants, including TEM-1 and TEM-2	For CATTTCCGTGTCGCCCTTATTC	22	0.4	800	13–34
	Rev CGTTCATCCATAGTTGCCTGAC	22	0.4		812–791
SHV-1 and variants	For AGCCGCTTGAGCAAATTAAAC	21	0.4	713	71–91
	Rev ATCCCGCAGTAAATCACCAC	21	0.4		783–763
OXA-1, OXA-4 and OXA-30	For GGCACCAGATTCAACTTTCAAG	22	0.4	564	201–222
	Rev GACCCCAAGTTTCCTGTAAGTG	22	0.4		764–743
CTX-M-1, CTX-M-3 and CTX-M-15	For TTAGGAAATGTGCCGCTGTA	20	0.4	688	61–80
	Rev CGATATCGTTGGTGGTACCAT	21	0.4		748–728
CMY-1, CMY-8, CMY-11, CMY-19 and MOX-1, MOX-2	For GCAACAACGACAATCCATCCT	21	0.2	895	3–23
	Rev GGGATAGGCGTASCTCTCCCAA	22	0.2		900–879
DHA-1 and DHA-2	For TGATGGCACAGCAGGATATTC	21	0.5	997	113–133
	Rev GCTTTGACTCTTTCGGTATTCG	22	0.5		1109–1088
VEB-1 to VEB-6	For CATTTCCCGATGCAAAGCGT	20	0.3	648	187–206
	Rev CGAAGTTTCTTTGGACTCTG	20	0.3		834–815
IMP variants except IMP-9, IMP-16, IMP-18, IMP-22 and IMP-25	For TGACACTCCATTTACAG	18	0.5	139	194–211
	Rev GATTGAGAATTAAGCCACCCT	21	0.5		332–313
KPC-1 to KPC-5	For CATTCAAGGGCTTTCTTGCTGC	22	0.2	538	209–230
	Rev ACGACGGCATAGTCATCATTTGC	20	0.2		746–272

^*^ Annealing position within the corresponding open reading frame (from the base A of start codon ATG).

**Table 2 foods-09-01726-t002:** Average number of Enterobacteriaceae colony-forming unit (CFU) count according to vegetable type.

Vegetable Type (*n* = 165)	Average log_10_ CFU/g Count (95% CI)
Bell pepper	5.36 (4–5.77)
Tomato	4.51 (2.6–5.51)
Cucumber	5.34 (5.17–5.49)
Radish	5.22 (4.52–5.48)
Spring onion	5.26 (4.9–5.53)
Gherkin	5,42 (5.1–5.5)
Carrot	5.51 (5.2–5.62)
Cabbage	5.36 (5–5.46)
Spinach	5.23 (4.9–5.32)
Parsley	5.48 (5.14–5.6)
Lettuce	5.36 (5.05–5.44)

**Table 3 foods-09-01726-t003:** Summary of the bacterial species isolated from different fresh vegetables, indicating the phenotypic resistance profiles and the genetic determinants detected.

Bacterial Species	Vegetable Sample	Store Type	Antibiotic Resistance Phenotype	Genetic Determinants
*Citrobacter brakii*	Carrot	Farmer market	AM, AMC, NA, CX, CTX	ND
*Enterobacter cloacae*	Carrot	Farmer market	AM, AMC, CX, CTR	ND
*Citrobacter freundii*	Carrot	Supermarket	AM, AMC, CAZ, CTR, CXM, MEM, IMI, GE, AK, CX, CTX	ND
^*^ *Serratia marcescens*	Spinach	Farmer market	AM, AMC, CAZ, AK, NA, FEP, CTX	*TEM, SHV*
^*^ *Morganella morganii*	Spinach	Farmer market	AM, AMC, CAZ, CXM, MEM, IMI, GE, AK, NA, CX, CTX	*KPC*
*Enterobacter cloacae*	Spinach	Supermarket	AM, AMC, CAZ, CTR, CXM, FEP, AK	*CTX-M*
^*^ *Escherichia coli*	Cucumber	Farmer market	AM, AMC, CAZ, CXM, GE, AK, FEP, DO	*CTX-M, TEM*
*Enterobacter cloacae*	Cucumber	Supermarket	AM, AMC, CXM, CX	*DHA*
^*^ *Enterobacter cloacae*	Parsley	Farmer market	AM, AMC, CAZ, CXM, MEM, IMI, GE, AK, NA, CX, FEP, CTX	*OXA-48*
^*^ *Klebsiella oxytoca*	Parsley	Supermarket	AM, AMC, CAZ, CTR, CXM, MEM, IMI, GE, AK, NA, CX, FEP, CTX	*KPC, SHV*
*Proteus vulgaris*	Lettuce	Farmer market	AM, AMC, CAZ, CTR, CXM, FEP, AK	*TEM, SHV*
*Enterobacter ludwigii*	Lettuce	Farmer market	AM, AMC, CX, CTX	ND
^*^ *Enteroacter cloacae*	Cabbage	Supermarket	AM, AMC, CXM, GE, COT, FEP, CTX	*CTX-M, SHV*

ND = Not determined. ^*^ Multiple Drug Resistant strain. AM = ampicillin, AMC = amoxillicin-clavulanic acid, NA = nalidixic acid, CX = cefoxitin, CTR = ceftriaxone, CXM = cefuroxime, CTX = cefotaxime, FEP = cefepime, AK = amikacin, GE = gentamycin, MEM = meropenem, IMI = imipenem, CIP = ciprofloxacin, LEV = levofloxacin, DO = doxycycline, COT = trimethoprim-sulfamethoxazole.

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
