# Peer review of "Prevalence of ESBL, AmpC and Carbapenemase-Producing Enterobacterales Isolated from Raw Vegetables Retailed in Romania"

_foods, 2020, doi:10.3390/foods9121726_

Round 1

Reviewer 1 Report

Line 201/202: Indicate whether CFU/g or cm2 or ml? Include in table legend.

Line 203: Include significance on the graph. 

Line 206: Revise this section.  There are no differences of relevance other than the tomato data. Again, you are reporting a 0.5 log or less difference.  That is not of relevance.  Why not discuss the difference associated with tomatoes. Are supermarket tomatoes washed and waxed and tomatoes at farmer market not.  What may account for the significant differences?

Line 230 – 236: Italicize genes here and throughout.

Line 307: Change CFU counts to population levels

Line 328: Revise this section. Microbial load on vegetables sourced from farmers markets and supermarkets were similar except for tomatoes.

Line 332/333: The table and in results section indicate tomatoes. Unclear at this point what information is correct.

Author Response

Point 1: Line 201/202: Indicate whether CFU/g or cm2 or ml? Include in table legend.

Response 1: log10 CFU/g has been added to the table

Point 2: Line 203: Include significance on the graph.

Response 2: The graph has been changed to signal the significance.

Point 3: Line 206: Revise this section.  There are no differences of relevance other than the tomato data. Again, you are reporting a 0.5 log or less difference.  That is not of relevance.  Why not discuss the difference associated with tomatoes. Are supermarket tomatoes washed and waxed and tomatoes at farmer market not.  What may account for the significant differences?

Response 3: We have revised and changed this section to:

"All in all, vegetable samples presented similar Enterobacteriaceae CFU counts whether they were acquired from supermarkets or farmer’s markets (Figure 2). No relevant statistical significance was observed, with the exception of tomato samples, which showed higher CFU counts in farmer markets (4.8 log) compared to supermarkets (2.6 log) (p<0.005)." 

We have also addressed the difference in tomato counts in the discussion. 

Point 4: Line 230 – 236: Italicize genes here and throughout

Response 4: Have been corrected

Point 5: Line 307: Change CFU counts to population levels

Response 5: We have modified accordingly

Point 6: Line 328: Revise this section. Microbial load on vegetables sourced from farmers markets and supermarkets were similar except for tomatoes.

Response 6: We have revised this section to clearly state that microbial load was similar between samples from farmer markets and supermarkets, except for tomatoes. We have also provided a possible explanation for the occurrence and added 2 references. 

"Vegetable samples displayed similar microbial loads whether they were acquired from supermarkets or farmer’s markets. The only exception were tomatoes, which displayed an average of 2.2 log decrease in CFU count in samples from supermarkets. This could be attributed to practices that are often employed by supermarkets in order to prolong shelf-life and increase the visual appeal of produce, such as thoroughly washing and applying edible coatings to produce. Tomatoes in retail markets are often waxed in order to prevent external microbial contamination, moisture desorption/absorption, all while maintaining their organoleptic properties for extended periods of time [56]. Furthermore, various substances that act as natural antimicrobials (such as essential oils) are being incorporated more and more in edible coatings [57]."

Point 7: Line 332/333: The table and in results section indicate tomatoes. Unclear at this point what information is correct.

Response 7: We have revised this section to clearly state that microbial load was similar between samples from farmer markets and supermarkets, except for tomatoes. We have also provided a possible explanation for the occurrence and added 2 references. 

Reviewer 2 Report

The manuscript has been significantly improved.

Author Response

Response: Thank you for your review. 

Reviewer 3 Report

It is still not clear for me, why the authors first incubate the samples, and then stsart counting.

This gives not a really count, as presented in fig. 2. The only thing you can mention here, that the isolatd were present on the 2g sample, however, in much lower quantities. 

I think it is necessary that the authors mention that. Moreover, during that enrichment it is possible that certain (groups) of mo will suppress the enterobacteriaceae. So, counting after an enrichment is nonsens. In this case it is probably done to isolate low numbers. That should be mentioned. 

Author Response

Point 1:It is still not clear for me, why the authors first incubate the samples, and then stsart counting.

This gives not a really count, as presented in fig. 2. The only thing you can mention here, that the isolatd were present on the 2g sample, however, in much lower quantities. 

I think it is necessary that the authors mention that. Moreover, during that enrichment it is possible that certain (groups) of mo will suppress the enterobacteriaceae. So, counting after an enrichment is nonsens. In this case it is probably done to isolate low numbers. That should be mentioned. 

Response 1:Peptone broth is a medium used for the growth of non-fastidious bacteria. The use of this broth followed by incubation of samples was done to achieve the main goal of the present article, which was to isolate and identify beta-lactamase-producing Enterobacterales. This was modeled taking into account previously used methods used by authors that isolated beta-lactamase-producing Enterobacteriaceae from vegetables. Therefore, we did not intend and did not use a method that would intentionally lower Enterobacteriaceae CFU count as it would greatly disrupt our primary focus. As it is stated, a 1:1000 dilution of the peptone enriched vegetable sample was performed and then streaked on EBM agar and incubated at 37 degrees Celsius for 24 hours. Without incubating, we would see no bacterial growth on the media. The broth sample was indeed incubated itself to encourage bacterial growth and allow isolation of those species present in low numbers, but only after a bit of the broth was taken for CFU count. 

This manuscript is a resubmission of an earlier submission. The following is a list of the peer review reports and author responses from that submission.

Round 1

Reviewer 1 Report

Below you will find the comments and remarks on the manuscript: 'Prevalence of ESBL ....'

Please explain the difference between Enterobacterales and Enterobacteriaceae. If there is no difference, use either one.

Maybe good to check the use of Itelics for bacterial names, also in the references

I would also put the results of the washing in the abstract

In the introduction the problem is mentioned: increase in resistant micro-organisms. But to substantiate this, more than 10 year old literature is often used. That is possible, but also provide recent literature, to show that the problem is still there. In the discussion you will find the new literature.

  1. 2, lines 68-75: Use EFSA data here to identify outbreaks
  2. 2, line 95: that's quite a lot, are potatoes included?
  3. 3, lines 121-129: I don't understand, why is incubation taking place? So you can get a very strange shift in groups, right?
  4. 3, lines 130-134: why EMB, and not an international medium for enteros (eg Violet Red Bile Glucose Agar)?
  5. 6, fig. 1: How do the authors know that the isolated strains are Gram-negative. They have not tested that.
  6. 6, fig. 2: is not legible to me, too small. The washing effect is minimal and comparable to other large trials (<90%); so don't call it "far more effective" here
  7. 7, fig. 3. It seems sensible to put these results on a log scale, then you can see more clearly that there is no relevant difference.
  8. 7, line 229: how are these strains isolated: from EMB ... then mention this
  9. 9, that is a hefty list, which does not invite you to read. Perhaps better to compare a few products in a table

Reviewer 2 Report

The aim of the study was to investigate the presence of β-lactamase producing Enterobacterales in fresh vegetables in Romania and if household washing is able to reduce their microbial load. It is a well-written paper, but similar studies on other regions, are available in the literature.

Although the limited novelty, significant data are presented.

Reviewer 3 Report

The paper addresses an important area, antibiotic resistant bacteria associated with fresh produce. The research is perhaps novel with respect to Romanian raw vegetables, but the literature is rich with publications on the subject. It is not clear why the investigators included in-home washing part of the study.  I suggest all the data and any narrative associated with that aspect of the paper be removed.  In general terms, less than a 1 log reduction in microbial population occurred following washing.  This is not new or of interest and has been demonstrated many times previously.

Specifically:

Line 23: Change Enterobacterales to enterobacteriaceae here and throughout

Line 45: Should be "antibiotic era"

Line 55: There remains no argument, fresh fruits and vegetables serve as vehicles for foodborne pathogens.

Line 68: Change human contamination to infection or carriage

Line 72: Delete and provide updated outbreak

Line 96: Many studies of investigated household washing. Typically, water is used as the control and house products including vinegar, lemon juice, etc are evaluated.

Line 131: What start with a 1:1000 dilution?

Line 207: What types of vegetables were included in the green salad

Line 220: Do not use percent difference should express as log 10 and then discuss as 1, 2, or 3 log reduction or difference. In food microbiology a 50% reduction may be meaningless...going from 2 x 105 to 1 x 105. This would be of little benefit from spoilage or pathogen perspective. 

L284: No clear what is meant by opportunistic pathogen. Salmonella is pathogenic to humans and animals, not plants.  So, are the authors indicating opportunistic to plants??